# The Beguine Option: A Persistent Past and a Promising Future of Christian Monasticism

**Evan B. Howard**

Department of Ministry, Fuller Theological Seminary 62421 Rabbit Trail, Montrose, CO 81403, USA; evanhoward@fuller.edu

**Abstract:** Since Herbert Grundmann's 1935 *Religious Movements in the Middle Ages*, interest in the Beguines has grown significantly. Yet we have struggled whether to call Beguines "religious" or not. My conviction is that the Beguines are one manifestation of an impulse found throughout Christian history to live a form of life that resembles Christian monasticism without founding institutions of religious life. It is this range of less institutional yet seriously committed forms of life that I am here calling the "Beguine Option." In my essay, I will sketch this "Beguine Option" in its varied expressions through Christian history. Having presented something of the persistent past of the Beguine Option, I will then present an introduction to forms of life exhibited in many of the expressions of what some have called "new monasticism" today, highlighting the similarities between movements in the past and new monastic movements in the present. Finally, I will suggest that the Christian Church would do well to foster the development of such communities in the future as I believe these forms of life hold much promise for manifesting and advancing the kingdom of God in our midst in a postmodern world.

**Keywords:** monasticism; Beguine; spiritual formation; intentional community; spirituality; religious life

---

## 1. Introduction

What might the future of monasticism look like? I start with three examples: two from the present and one from the past. My first example is Jessica and Weston who, along with their son Simeon, own a home in Minneapolis, Minnesota in a neighborhood of amazing diversity and some pretty serious need. Weston is a carpenter and makes some of his income building things. But they also raise a little support through a Christian non-profit so they can devote time to art therapy, God-discussions while doing woodwork with neighbors, and other creative ideas. They live near friends who share a common Christian vision and they pray with them regularly. Some neighbors are associated with the same non-profit, an organization which asks that they make a serious effort to blend contemplative, prophetic and missional currents of Christianity in their life and work. Both Weston and Jess meet with spiritual directors who provide support and accountability. Sometimes they struggle to see just where they fit in all of this but they are willing to play with it.

My second example is not an individual, but a group—The Church of the Sojourners in San Francisco. This community started in the 1980s with a few families who wanted to share more deeply the life of Acts 2. Singles and families have various jobs but pool most of their money together into a common fund which pays for their shared housing, utilities, transportation and other common expenses. They worship together on Sundays and eat together frequently. Important decisions are made in common. They often absorb into their community a few people who we might call higher need. They share a covenant of commitment to Christian values and while not requiring involvement

in any particular "outreach program," this community has manifest, for many years, the presence of Christ in their neighborhood.

My final example (from the past) is Marie d'Oignies, the woman some identify as the "first Beguine" (see especially King and Feiss 1993; Mulder-Bakker 2006). Marie grew up in the town of Nivelles in the diocese of Liège near Brussels in Belgium. From her childhood, she was attracted to simplicity and refused the fine clothes her parents offered her. Her parents arranged that she would be married at age fourteen to John, the son of another wealthy local family. After her marriage, she began to practice the ascetical life more seriously. John also came to a deeper relationship with God and they moved to nearby Willambroux and practiced a semi-monastic life together: fasting and spending their time in prayers, physical labour, and works of charity. She and John participated with an informal community near a group of lepers. John disappears from the story but we hear of Marie's popularity growing exponentially. She moved once again in 1207 to a cell in Oignies to avoid attention. There she maintained her practice of prayer and provided spiritual counsel. She died in 1213 and her example stimulated a host of women's communities.

In 2017, Rod Dreher, senior editor at *The American Conservative*, published *The Benedict Option: A Strategy for Christians in a Post-Christian Nation*. This book, which became a much discussed best-seller, advertises itself as "a guide for Christians under siege today" (Dreher 2018, back cover). "Recognizing the toxins of modern secularism, as well as the fragmentation caused by relativism," Dreher urges, "Benedict Option Christians look to Scripture and to Benedict's Rule for ways to cultivate practices and communities" (Dreher 2018, p. 18). Dreher's vision is to facilitate a visible expression of philosopher Alasdair MacIntyre's call for a "new Benedict": a figure who symbolizes the foundation of religious communities whose lives together nurture the development of virtue (MacIntyre 1981). Dreher's call to a "Benedict Option" is not the first proposal for alternative communities of virtue. Indeed, the phrase "new monasticism," as used in the past fifteen years, was coined from a similar desire to respond to MacIntyre's call for a "new Benedict" (Wilson 2010; House 2005). I applaud these proposals for creative communities of virtue. I wonder about the link to *Benedict*.

I have been visiting both "new monastic" communities and more traditional congregations of religious life for over thirty-five years. As I visit these communities and read their own self-evaluations, I am struck with a conviction regarding both traditional religious life and "new monasticism" that while formal "monastic" entities (institutes of religious life—for my use of monastic and other terminology, see Howard 2008, 2013), are vital and will remain for perpetuity, a promising future for religious life in the next decades may be discovered in less institutional forms of life. Those who live these forms of life may not make solemn vows. Their communities may not be recognized by a formal ecclesiastical body. Yet their values, their rhythm of activities, and their sense of community and identity all bear sufficient resemblance to recognized religious institutes that it is appropriate to identify these forms of life as a semi-religious movement. Kaspar Elm identifies such expressions as *vitae regularis sine regula*, a "regular" or religious life without a formal rule (Elm 2016).

One example of this kind of less-institutional expression of devotion in history is the varied communities of women (and men) known in Europe especially in the thirteenth and fourteenth centuries often identified as "beguines" (the communities of men were called "beghards"). Both contemporaries and scholars universally interpret them as "semi-religious" (McDonnell 1969; Simons 2001; Dean 2008; Swan 2014). Indeed, they are often identified as a type of semi-religious life (Elm 2016, pp. 286–87). Thus, while there have been a number of less-institutional expressions of lay devotion in Christian history, the beguines, in all their ambiguity, function nicely as a counter-symbol to the institutional image of Benedict. Furthermore, just as Dreher and others have turned to the context of Benedict for parallels to this present age, so I think we can identify instructive parallels between the contexts of beguine experiments and contemporary search for appropriate religious life, indeed, perhaps moreso. For these reasons, I will refer to expressions which seek to live an ordered life of devotion yet without formal institutional affiliation or structure (intentional but not as institutional) as "The Beguine Option."



My aim in this essay is thus to propose a fresh mythological support for the ideological reframing of religious virtuosity as expressed in Catholic religious life and Protestant new monastic expressions today (Wittberg 1994). Wittberg argues that the ideological frame of traditional Catholic religious life has collapsed and that "the most commonly suggested replacement" "was unable to be realized on a communal basis" (Wittberg 1994, p. 240). At this same moment, Protestants are experimenting with religious life, open to ideological suggestions (House 2005; Graham Cray and Kennedy 2010). There are currently groups founding twenty-first century "beguine" communities today (Dean 2008, Raber 2009). Thus first, I see indications that less-institutional, ecumenical experiments in semi-religious life could be profitable in the decades to come (at least), and second that the image of a new *Beguine* Option may just have the mythological strength to support this fresh ideological frame.

I would like to suggest, in the present article, that this Beguine Option—expressions which seek less institutional forms of whole-hearted devotion—can be identified throughout the history of Christianity. Furthermore, this Beguine Option describes (with some points of divergence that I will develop later) a wide range of contemporary experiments in religious life, experiments which I believe are uniquely suited to meet the needs of a postmodern generation. In what follows, I will first present a description of beguine life in the in the thirteenth and fourteenth centuries as summarized from the diverse communities identified with the name. I will then turn back to identify similar expressions found throughout Christian history, demonstrating the *persistent past* of the Beguine Option. Moving from the past to the present, I will next describe a variety of contemporary semi-religious expressions, noting the significant similarities and differences from the Beguine Option as already described. Finally, I will place my discussion in dialogue with other reflections on the future of monasticism, suggesting that some kind of Beguine Option has unique promise for living and spreading the Gospel of Christ in a postmodern generation.

## 2. The Medieval Beguines

Describing beguine life is complicated by a number of factors (Dean 2008). Scholars struggle to identify the meaning of the term "beguine" itself (McDonnell 1969, pp. 430–38; Simons 2001, pp. 24–32, pp. 121–23; Miller 2007). The term was used both as attribution of praise and as a term of derision, with the consequence that historians today examining uses of the term must be careful identifying the links between terms and communities (Böhringer et al. 2014; More 2018). "Beguine" also was just one of a number of terms used to describe similar informal women's communities throughout Europe (Dean 2008; Swan 2014, p. 12). Furthermore, some women who eventually joined more institutional orders (for example, Cistercians or Third Order mendicant orders) still pursued values and practices associated with their non-institutional counterparts (Lester 2011; More 2018). Fourth, sources for the documentation of beguine life are hard to find and "house rules and statutes have been neglected in the literature" (Böhringer et al. 2014, p. 224; see de Vries 2016). Finally, beguine communities were independent and practices varied between groups. Thus, any summary of "beguine life" must be seen only as a rough composite drawn from the most common practices that can be identified from the sources available.

One searches for a single "founder" of the beguine movement in vain. It is best to see the rise of beguine communities (however they may have been named) as responses to changes in the medieval world, changes that parallel developments in the post-modern world today. Europe was experiencing something of a renewal following what has been termed "the twelfth-century renaissance" (Benson et al. 1982). Cities and roads were being developed. Universities were gaining strength. A new economy based on coinage was emerging. At the same time, new vehicles of production and consumption were developing: forest clearing, cloth merchants, fairs or festivals, credit instruments and more. Some call it the emergence of a "profit economy," an economy that gave birth to new rich and new poor alike (Little 1978; Wood 2002; Ekelund et al. 2015). The new rich were the merchants, who were able to link producers and processes of production with consumers. Some religious movements—such as Francis of Assisi and his earliest followers—responded to these developments with an outright

rejection of money and the socio-political stratification associated with these developments (Flood 2001; Flood 2010; Vauchez 2012). Some beguines, however, explored these developments as opportunities for community independence, for example through involvement in the cloth industry (Simons 2001, pp. 8–10; Miller 2014a, pp. 59–80). The Gregorian reforms and land acquisitions brought new power to the Church. The world of monasticism had become something of a smorgasbord of competing options, each religious order claiming to be the clearest expression of the apostolic life. At the same time, pious laity were finding new avenues of devotion as some sought the apostolic life as hermits or as traveling preachers, while others formed confraternities of mutual support (Vauchez 1993; Melville 2016, pp. 89–124). Ecclesiastical opinion of these new avenues varied. The communities of devout women we call beguines (or other kindred terms) were formed in the midst of this shifting range of possibilities.

The situation for women was exceptionally complex (Bynum 1987). Alison More writes, "Both men and women sought perfection in the world; however, the social and secular forms of devotion that emerged were more problematic for women and their male contemporaries. In particular the lay piety performed by men was often incorporated into guilds, confraternities, or other structures that were seen as having a distinct secular role. In contrast, there was no acceptable outlet for some to carry out public devotion" (More 2018, p. 4; also Dean 2008). On the one hand, women's monastic houses or double monasteries of men and women were being established in greater numbers than ever before (some by women—see, for example, Lester 2011). There was also the option—particularly in the later twelfth through fourteenth centuries—of joining various "third orders," particularly in association with a recognized mendicant group. Indeed, narratives of devout women periodically highlighted (or overstated, or invented) associations with recognized orders in order to protect women whose way of life was perhaps more independent and less institutional than some might approve (More 2018). But a third option was to explore a less institutional route, founding informal communities of like-minded devout. The environment was ready for women who wanted to make a life for themselves in pursuit of Christ to try something new. It is from this milieu that the beguines emerged.

The process of founding a beguine community varied greatly (for examples, see Simons 2001, 36–48; Miller 2014a; research is also summarized in Swan 2014, pp. 23–48). At times, someone was able to procure a home (or a sympathetic person would endow the home) and sisters would move in and set up a community. Sometimes, a group of devout women (or men—beghards) would cluster around a "holy person" and a community would form, either living in their own homes and meeting in common or finding a place of common residence. At times, a group would follow a preacher who would then help establish this following into an intentional community of semi-religious devout. Some communities developed a close relationship with a local church while others were fiercely independent. Thus, the housing arrangements of beguines varied according to the size and wealth of the community. Beguines could live in private homes or in a communal dormitory. The layout of a communal estate itself varied. Some beguinages had no chapel of their own and the residences were not necessarily "planned" for the beguinage. These were called "convent" beguinages. Other "court" beguinages were developed with a main central church and a courtyard around which the residences were arranged (Simons 2001, pp. 48–60).

Other elements were important in founding a beguine community. If the members were to live within a common house, they needed to secure potential sources of income. While a few, such as the community gathered around Ida of Nivelles, practiced begging, most beguines either supported themselves and the community through independent resources or through manual labor. Another factor involved was the identification of a common vision, often a longing to live in the mix of active and contemplative lives, where the pursuit of God in prayer and an active life through charity, manual labor, and teaching were blended in different ways within each community (Simons 2001, pp. 61–90; Miller 2014a, pp. 35–80; research also summarized in Swan 2014, pp. 71–83). Vision is, in turn, connected to motive. Different people might find beguine membership attractive for different reasons. Tanya Stabler Miller writes:

Wealthier women might join a beguinage in pursuit of apostolic ideals. They also might find the beguine life a more flexible, and thus more appealing alternative to the cloister, where they would be cut off from family, friends, and associates. At the same time, beguinages offered women of middling and lower socioeconomic status a safe haven in which to worship and work (Miller 2014a, p. 36).

One more element in the foundation—or at least the preservation—of a beguine community was permission. Permission was needed by many people for many things: permission to preach or encourage one another, to celebrate mass together, to form a business, to dress in ways that looked like a nun, to waive one's obligations in military or legal services (involving the use of oaths), and more. For this reason, it was advantageous to have associations with people in high places who might plead your cause. Perhaps the most celebrated examples of this are Jacques de Vitry who advocated for the beguines of the Liège region before Innocent III in 1216 (for various assessments, see Sandor 1988; Neel 1989; Wright 2012) and King Louis IX of France who founded and supported the royal beguinage in Paris (see Miller 2014a, pp. 14–34).

As I mentioned above, a summary of the "life of a Beguine" can only be a composite drawn from a variety of sources. Some works present the life of a single beguinage and others draw from the records of a range of communities. My aim here is simply to present a broad summary, drawing from a couple of studies (de Vries 2016; Miller 2014a, pp. 37–41; research also summarized in Swan 2014) to demonstrate that beguines perceived themselves as attempting to lead a "religious" life—an intentional form of life centered around Christian devotion—while avoiding the institutional connections.

Evidence for regular common meetings abound in the literature, though what meetings were common in what communities varied. Common meetings often included mass, community prayers, gatherings to conduct the business of the community, and selected meals (Miller 2014a, pp. 105–7). These are the normal kinds of common gatherings one expects to find in monastic environments. Historian Jennifer Deane underscored the importance of their times of prayer, recounting that, "beguines were petitioned or even required in statutes to pray for their benefactors, and in some communities the daily prayer cycle was structured along lines similar to monastic schedules" (Dean 2008).

In many houses, times of work were both common and times of prayer. One can perceive the blend of prayer and work in this account of the Beguinage of Saint Elizabeth in Ghent:

> On work days they hold to the practice of rising early in the morning and coming together in the church, each going to her own place, which she has specially assigned to her, so that the absence of anyone may thereby be more easily noticed. After they have heard the Mass and said their prayers there, they return to their houses, working all day in silence, in which thing they are considered very useful to the whole country. And while working thus, they do not cease from prayer, for in each convent the two women who are best suited for this recite clearly the psalm "Miserere" and other psalms which they know, and the "Ave Maria," one singing one verse, the other the next, and the rest silently with them, or diligently listen to those who are reciting. Late at night, after Vespers, they go into the church, devoting themselves to prayers and meditations, until the signal is given and they go to bed. On Sundays and holy days, with masses and sermons, prayers and meditations, they devote themselves to the Lord's service in all things; nor may anyone leave the Beguinage on these days without special permission from the principal mistress. (Amt 2010, p. 214)

The quote above reflects a beguinage at which their primary work is performed within the walls of the convent itself. Yet quite often beguines went out to serve in hospitals, leprosariums, schools or other similar places. Indeed, activity, whether in employment or in service, was a distinctive character of beguine forms of life. Yet even as they left their domicile, care was taken through specific rules to preserve an element of monastic "separation" from the world and to guard the reputation of the women who served outside the beguinage (de Vries 2016).

Though beguines did not generally renounce their possessions nor make formal vows of poverty (indeed, some beguines were shrewd merchants: managing estates, administrating segments of the textile industry, and granting loans), they heartily embraced a life of simplicity (de Vitry 1998, pp. 81–82; Simons 2001, pp. 62–68; More 2018, pp. 22–24). The *Règle of Perfect Lovers*, a late thirteenth-century devotional treatise designed to describe beguine virtue, includes poverty as one of the four pillars of beguine life (along with purity, humility, and love. Miller 2014a, p. 113). After Pope Nicholas's decree *Supra montem* in 1289 which established a more formal "Franciscan" third order, informal beguines were often directed (or forced) to join this expression, where they could still hold property, yet make commitments to simplicity (More 2018, pp. 37–38). Similarly, with regard to chastity, beguines did not take formal vows of chastity—and were free to leave their beguinage in order to get married—yet the virtue of chastity was highly praised among the beguines and their admirers (de Vitry 1998, p. 52; Miller 2014a, p. 16).

Prospective members would join a community through a combination of availability, existing relationships, and formal application. Women might be permitted to join for a trial period of one year, after which there was a welcoming ceremony. Beguines generally did not make vows of stability and so were free to leave to get married or for other reasons, though it appears that house commitments generally lasted. While ministry activities and church attendance required that beguines would leave the beguinage on occasion, the kinds and amount of allowable travel was regulated (de Vries 2016; Miller 2014a, p. 38), although the beguines in Paris maintained a lively network with patrons, spiritual advisors, and theologians of the region (Miller 2014a, p. 40).

The clothing of a beguine was designed by each beguinage, often sewn by their own members with their own cloth, strictly avoiding decorative embellishments. The regulations at Ghent declared that "none may have anything which is unusual or suspect in its shape, sewing, or belting, or in the way of nightcaps, hoods, gloves, mitts, straps, purses and knives" (Amt 2010, p. 215). Miller describes their dress as "humble garb evocative of habits" (Miller 2014a, p. 1). The community itself was generally led by a *magistra* or mistress, chosen by the community. The magistra supervised the affairs of the community and could also serve as the in-house spiritual director. Most beguinages lived according to a rule of life and yet this rule was not an officially recognized document but was "the living tradition of a particular beguine community that had been formulated by the women themselves" (Swan 2014, p. 59), much as customaries functioned in earlier monastic communities. Thus, without formal habit, abbess, or approved Rule of life, the beguines dressed and ruled themselves much as a monastic community.

As one can imagine, not everything was harmonious and pure in beguinages. It is likely that some (though certainly not all) of the accusations of sexual impropriety by women who were not cloistered actually happened (Swan 2014, pp. 64–65). Hadewijch, beguine visionary and mystical writer, was likely evicted from her beguinage (Hart 1980, p. 4). There is evidence that beguinages, just as with traditional convents, developed means by which members were corrected and conflict was resolved (Amt 2010, p. 215).

As we have seen, though the women in beguinages established a common life of devotion together, based on commonly held values, they did not make formal vows. Historian Gert Melville describes this element as characteristic of beguine life: "The movement belonged to women who wanted to live a life of poverty, penance, humility, chastity, and deeply interior piety but who had no intention of professing vows, observing a traditional rule, or retreating from the world into a monastery" (Melville 2016, p. 193). The 'rules' established by the headmistress of the beguinage in 's-Hertogenbosch stipulated that beguines should be obedient to their headmistress (see de Vries 2016, p. 137), but this is not to be equated with the vow of obedience in traditional monastic settings. Intentional, but not as institutional: this is characteristic of beguine life. We have seen this above with reference to nearly every area of life: clothing, housing, work, leadership, and on and on. The beguines sought to live a fully devout life and yet also chose to do so in conscious distance from the formal mechanisms of religious life. *Vita regularis sine regula* (Elm 2016).

And it is really their devout life that is most important. Jennifer Deane warns the interpreter of beguine life of being fixated on what the beguine is *not* (not married, not nun, and so on. Dean 2008). Robert of Sorbon, in the mid-thirteenth century, urged the students at the new college he was establishing in Paris, to look to beguines as models. Miller writes, "Robert associated the label 'beguine' with a specific set of behaviours compatible with his goals for his college: a community of secular clerics committed to living a religious life defined by what one *did* rather than by a particular habit or rule" (Miller 2014b, p. 142). When we turn our attention from what the beguine was not toward what they *were*—how they spent their lives—we discover the value of this choice of life in their context. For the beguine, life was a life of community self-government, of economic participation in the market, of a blend of contemplative mysticism and apostolic service to those in need (Simons 2001, pp. 61–90), and of educational development (Miller 2014a, pp. 103–25), a form of life that was extremely difficult for women of that time to realize.

Furthermore, in spite of the many ecclesiastical tensions surrounding beguine life, the beguines succeeded as a movement. Walter Simons reports the foundation of two hundred beguinages before 1320, supporting several thousand beguines (Simons 2001, p. 109). He explains this success by the blend of the contemplative and active offered through the beguine form of life. I would suggest that the offering of this blend was possible in medieval culture uniquely (and perhaps iconically) through the intentional, but not institutional way modeled by the beguines. Walter Simons claims as much in his overview of Christians "on the margins of religious life: hermits and recluses, penitents and tertiaries, beguines and beghards." After noting the unique feature of beguine/beghard movement as consisting almost entirely of women, he concludes his review of the various medieval marginal expressions by citing Jacques de Vitry who proclaimed that:

> We do not consider religious only those who renounce the world and go over to a religious life, but we can also call regulars all the faithful of Christ who serve the Lord under the evangelical rule and who live in an orderly way under the one highest and supreme Abbot. (Simons 2014, p. 323)

Similarly, Gert Melville concludes, "the Beguine way of life remained powerful enough, because of both its strength in numbers and the depth of its faith, to continue to present an alternative path to the monastery's traditional purpose: preparing the way for the individual soul's journey back to God" (Melville 2016, p. 195). I concur with de Vitry and Melville. Furthermore, I find this way of life to be a worthy model for the future of religious life today. It is the forms of life resembling these late medieval devout women that I am calling the Beguine Option.

## 3. A History of the Beguine Option—The Persistent Past

Devout Christians who desired to live in common, but not necessarily to be enclosed in a cloister; believers who wished for local permission to pursue their way of life, but did not need to go the route of formal papal recognition; sisters and brothers who made sincere promises to live a life of simplicity, purity and mutual submission, but who were not prepared to renounce all property and make formal public vows, Christians who wanted to live as religious, but did not require a developed Rule of Life—this intentional but somewhat less institutional Beguine Option has a long history in the Christian Church, a history worth remembering. My survey of this material will be both sweeping and yet selective. My aim here is simply to demonstrate that the Beguine Option is not an isolated phenomenon in history. Similar forms of life have appeared in various times and places throughout Christian history.

### 3.1. The Patristic Period

Indeed, current scholarship regarding the pre-history of Christian monasticism is revising the simple "Antony—founder of anchorite institution; Pachomius—founder of cenobite institution" model of monastic origins in light of evidence pointing toward early informal local collections of ascetics or

*apotaktikoi* who led a life of strict religious practice, served local villages, but did not establish formal institutions (see Judge 1977; Goehring 1999; Harmless 2004, pp. 417–23). We also find Christians, very early, like Justina (d. 268) who "began monastic observance in her home, cut her hair in the common style of monastics, and observed the seven daily periods of prayer" (Swan 2001, p. 112; see also p. 123). The domestic ascetic movement, a movement which itself was probably connected with that of widows and virgins (Clark 1986; Elm 1994), was influential in the foundation of monasticism throughout the Roman Empire (Palladius 1964, pp. 142–43; Silvas 2005, pp. 75–83; Theodoret of Cyrrhus 1985, pp. 186–89; Magnani 2018).

Palladius' fifth century *Lausiac History* documents a wide range of forms of life identified with the term "monk" or "monastic life" (e.g., Palladius 1964, pp. 49–51). In the early Christian east, a common expression of religious life was the wandering ascetic, following Jesus's own pattern of itinerant ministry. Robert Murray, in his article on "The Features of the Earliest Christian Asceticism," writes of these early examples of devout life, "Certainly the wandering ascetics of the Syriac-speaking Near East resisted all institutionalization" (Murray 1975, p. 70). Rather than understanding the origins of Christian monasticism in terms of a few founders who established recognized institutions, it is probably best to see writers like Athanasius and Jerome drawing attention to certain forms of life in the midst of many others, certain forms which then became the dominant symbols (and approved institutions) of Christian monasticism (Harmless 2004, pp. 417–58). At the council of Chalcedon, the ecclesial hierarchy deemed it necessary to require monasteries to be placed under the direct supervision of the local bishop and "only the bishop's express authority could allow new foundations" (Melville 2016, p. 10). The ruling was not enforced and informal expressions appeared throughout Christiandom. Perhaps the importance of this regulation was that officials would periodically appeal to it when they wished to curb particular expressions.

### 3.2. The Medieval Period

As we move into the medieval period, I will only provide examples from the Christian West, as the developments in Coptic and Byzantine monasticism are complex, my knowledge here is limited, and the sources are less available (see Morris 1995; Hatlie 2007; Vivian 2002). We find a description of domestic ascetic expressions in Patrick of Ireland's writings. Westley Follett writes:

> In his *Confessio* Patrick marvels at the conversion of people who had only recently worshiped idols and that the sons of the Irish and the daughters of their kings had become 'monks and virgins of Christ'. While it is doubtful that we should understand these as cenobitic monks and nuns in a Pachomian or Benedictine sense, there is no question from Patrick's remarks in the *Confessio* and his *Epistola ad milites Corotici* that the promotion of the religious life, and particularly its commitment to celibacy, was a central feature of his ministry to the Irish. It seems likely that Patrick's virgins remained at home with their unbelieving families rather than forming a separate Christian community. (Follett 2006, p. 28)

Our understanding of Celtic monasticism has matured. We no longer see Celtic "monasteries" in light of contemporary models, but rather in terms of the settlements early Celtic religious life resembled. Part of the discussion around the nature of Celtic monasticism is the identification of the *manaig*. *Manaig* have been considered as tenants of monastic properties, yet they were also persons or family groups who lived within range of the monastic enclosure and shared some of the benefits of the community life and rhythm, and were also obliged to keep some of the ritual and monastic practice. We see something similar in the early *minsters* of England. Sarah Foot writes, "The household of a widow living in quiet seclusion with her unmarried daughters might be thought of as a *monasterium*, just as was a new community created by the royal grant of a portion of land to an aspiring abbot and a group of like-minded men." (Foot 2006, p. 5).

Following the example of Frankish queen Radegundis (Melville 2016, p. 16), many monasteries were founded by nobles such that Marilyn Dunn describes Frankish monasticism of the seventh

century as a situation "where monasteries directed by family groups became centres of aristocratic power as well as religious cult" (Dunn 2003, p. 158; for the Columbanian tradition, see especially Fox 2014). This kind of expression is not that different from the life of Macrina and Basil in fourth century Cappadocia. The synods of Aachen (816–819) attempted, with mixed success, to unify all monastic communities within the Benedictine rule and to encourage priests to become more "monastic" through commitments to common residence and a common rhythm of prayer (Melville 2016, pp. 40–42). Yet even as the Benedictine flowering reached its peak in the years of Clunaic dominance, many hermits would find their way near the monastery to live a loosely institutional but highly intentional life of devotion (Leyser 1984, p. 19).

In the eleventh century, we hear of the Hirsau reform movement through which many lay people "gathered together to live the 'common life in the form of the primitive Church'" Pope Urban II gave permission for this movement in a bull of 1091. Converts renounced life in the world, living together without entering a religious house, but rather dwelling in pious communities (Grundmann 1995, p. 223). Similarly, in 1122 Norbert of Xanten urged his convert Theobald of Champagne not to join the Praemonstratensians, but rather to marry and live a religious life in the world. Thus, Norbert and Theobald formed the *Fratres et Sorores ad Succurrendum*, a third order which "anticipated by a century the Franciscan Tertiaries" (Davison 1927, p. 91).

All of these expressions bear similarity to—and can serve as early historical examples of—the Beguine Option I descried above. Now we are up to the period of time when the beguines themselves emerged. As mentioned in the section on the beguines, the twelfth century was a period of lay flowering. Lay preachers proclaimed the faith in the town squares. Confraternities of laity formed together and committed to regular times of support, common service activities and even standards of holy living (see Vauchez 1993, pp. 107–27 and Thompson 2005, pp. 69–102). Ultimately, in 1221 the Brothers and Sisters of Penance received official recognition through a *Memoriale propositi*. Another example of a similar expression at this time is the Humiliati. Frances Andrews, author of the first English-language major study of this group, summarizes the earliest references to the Humiliati as describing "both groups of clerics living in community and lay men and women devoted to the religious life in small ad hoc associations promoting the catholic faith" (Andrews 1999, p. 2; see also Brasher 2003; Brasher 2017). Condemned as heretics in 1184, they appealed to Rome and were ultimately accepted in 1201 in part through the creation of a Third Order of married Christians eager to pursue the devout life and even to encourage their neighbors through preaching. After recounting the story of the Humiliati and their acceptance, Gert Melville proclaims, "The church had blazed a new trail. It had begun to occupy a new social space, with the help of the forces that that same space had produced" (Melville 2016, p. 200).

The trail that was blazed by the approval of the Humiliati was widened into a road—and later transformed into a highway—through the incorporation of the community of Francis of Assisi and other mendicants (on Francis see Moorman 1988; Thompson 2012; Vauchez 2012; removed for peer-review, chapter 2; on Domincans see Hinnebusch 1966; for other mendicants see Andrews 2006). Francis chose to leave Assisi and the merchant option he had available. Nor did he join any of the local monasteries. At the start of his ministry, Francis of Assisi chose to live as a lay penitent, living and serving lepers, much as Marie d'Oignies and her husband did in Liège in these very years. In time, the little brothers of Assisi received approval from Rome not only as a community of penitents but also as an international agency of gospel proclamation (as did other mendicant orders during this period). The Order of Friars Minor, even within the final decade of Francis's life, struggled to navigate the uneasy tensions between radical abandonment to God, apostolic service, voluntary poverty, and faithfulness to the church. Over the course of the thirteenth century—and not without its share of internal conflict (Burr 2001)—the Franciscans institutionalized in three orders as a mission arm of the Roman church (on the institutionalization of the second order see Knox 2000; More 2018). The third order, along with the third orders of other mendicant orders served as officially recognized religious institutes with whom lay devout who desired or needed might choose to affiliate.

Other informal intentional groups, such as the Poor Catholics, also received a measure of approval in this same window of time. Nonetheless, the thirteenth and fourteenth centuries were not only periods of high interest in the semi-religious life but also mixed approvals of the same. It is fruitless here to document the changing fortunes of those who sought to live a *vita regularis sine regula* (Elm 2016; see Grundmann 1995). André Vauchez describes the fate of the penitential confraternities as follows:

> The number of penitents, both isolated or in communities, remained large until the end of the Middle Ages. But beginning with the early fourteenth century, the papacy tried to channel this kind of religious life, which was popular with both laymen and laywomen, in the direction of greater regularity. The confraternities of penitents were absorbed by third orders, which recruited their own members primarily among women, and which themselves often evolved toward claustral forms of life, ending up as semi-monastic congregations. In this way the originality of this typically medieval movement was progressively lost—though the movement was to be reborn in new forms in the sixteenth and seventeenth centuries. (Vauchez 1993, p. 127; yet also see More 2018)

One final example may serve to fill out our sampling of the Beguine Option through the middle ages. John Van Engen writes of the Sisters and Brothers of the Common Life:

> Beginning in the 1380s, in market towns along the IJsel River (east-central Netherlands) and in the country of Holland, groups of women and men formed households organized as communes and a lifestyle centered on devotion. They lived on city streets alongside urban neighbors, managed properties and rents in common, and prepared textiles or books for local markets–all the while refusing to profess vows as religious or to acquire spouses and property as lay citizens. (Van Engen 2008, loc 99)

The Sisters and Brothers of the Common Life thrived in the fourteenth and fifteenth centuries, in part due to their legal expertise in navigating canonical privileges. Today we are familiar with this group because one of their number, Thomas à Kempis, penned the influential *Imitation of Christ*. Some of these modern devout, as they were also called, formerly joined or formed houses of canons regular, most notably the canons of Windesheim. Yet many other devout remained in households ordered through informally written "ways" or "customaries" (Van Engen 1988, pp. 155–86). Kaspar Elm describes the self-conscious awareness of the modern devout precisely as a semi-religious expression:

> The early Christian hermits of Egypt, Syria and Palestine, Elijah and Elisha, the prophets of the Old Testament and the pious Essenes were for them not merely the prototypes of monks and monasticism, but creators of free societates that lived sine regula, sine statutis obedientialibus, sine habitu aut ceremoniis regularibus. ... What monks, canons, and mendicants, and indeed the military orders had long claimed for themselves—to live according to the model of the Apostles and to return to the early church—semi-religious now claimed as well, and with an exclusivity similar to that of other orders and their branches. (Elm 2016, p. 308).

### 3.3. The Modern Period

In the modern period, the Western Church divided and the options for semi-religious life in time became more available to all the divisions. Indeed, it is not really fair to talk about "intentional yet less institutional" options, because after 1540 or so, the institutions themselves begin to change, enabling opportunities that were not available in the medieval period.

Approval of the Society of Jesus (the Jesuits) as a religious order did not come easy (Meissner 1992, 178–181). After much discussion, the band of Ignatius of Loyola's followers submitted their proposal for approval which was received, at first. Contrary voices complicated matters in the finalization process, however. Some were concerned about the Jesuit abolition of recitation of the divine office in choir. But more central to the final judge was the creation of a new religious order itself. After much prayer (and perhaps a little human persuasion), the Society of Jesus was finally confirmed on 27

September 1540. This was a significant development in that (1) the Vatican approved the creation of a new order, and more significantly that (2) this order was designed—perhaps even more than the Dominicans—as a vehicle for apostolic ministry.

Because they were a dispersed, missionary society many of the features of traditional monastic life had to be revised or abandoned. Jesuits could or should not be called back from their ministries to join in the communal recitation of the divine office or attend chapter meetings (Taft 1993, pp. 301–2). The requirements associated with these were released. Often, they would form houses within which a team of missionaries would be based. Those who were present might gather regularly for meals and prayer. Daily private self-examination and regular confession and reception of the sacrament was expected. But this was an army at war. And because of this, adjustments were made. As one of their early leaders proclaimed: "We are not monks; the world is our house!"

The next step came thirty-five years later. The Roman church had issued a clarion call for evangelism and purity of life through the council of Trent (1545–1563), and Philip Neri (1515–1595) embodied the spirit of that call superbly. He spent his time as a layman and as a tutor, ministering among those without education, without resources, or those in the prostitution industry. He helped found the Confraternity of the Most Holy Trinity of Pilgrims and Convalescents to minister to the needs of poor pilgrims. He initiated regular gatherings for preaching, song, and discussion and pioneered many creative means of communicating the Christian faith. In time, he was ordained and it seemed fit for him to organize a community of secular priests in order to provide mutual support in the work. This was the beginning of the "Congregation of the Oratory of St. Philip Neri (or Oratorians. see Moss 1957, pp. 166–67; http://www.newadvent.org/cathen/11272a.htm)." This community, though with pontifical approval, chose not to make solemn vows to a large order, but rather to offer simple commitments to membership within local, self-governing houses or Oratories. A Rule of Life was drawn up for the Oratorians seventeen years after Neri's death as a collection of the custom of Neri and his followers. Philip Neri is commonly thought of as the "father" of the Societies of Apostolic Life, religious groups that do not make formal religious vows, may own property but usually live in some arrangement of a supportive community. Societies of Apostolic Life are usually organized around particular tasks or mission goals.

The third step was made through the combined work of Vincent de Paul and Louise de Marillac. Vincent de Paul helped start the Confraternities of Charity around 1621, which volunteered to care for sick, beggars, victims of war and others in similar straits. The ministries of these charities grew as both volunteers and generous gifts multiplied. But they needed administration and a common vision. Together, Vincent de Paul and Louise de Marillac were the parents of the Daughters of Charity. Louise de Marillac began to visit the various ministries in 1629. She studied their activities, corrected abuses, and revived their zeal. There were a number of younger women serving the charities and she held these girls in her heart. She was burdened for their spiritual formation. Thus, in 1633 de Paul and Marillac formed the Daughters of Charity, gaining formal approval for their life and Rule in 1645. Here is how the second paragraph of this rule begins:

> They should consider that although they do not belong to a religious order, that state not being compatible with the duties of their vocation, yet as they are much more exposed to the world than nuns—their monastery being generally no other than the abode of the sick; their cell a hired room; their chapel, the parish church; their cloister, the public streets or the wards of hospitals—they are obliged on this account to lead as virtuous a life as if they were professed in a religious order; to conduct themselves wherever they mingle with the world with as much recollection, purity of heart and body, detachment from creatures; and to give as much edification as nuns in the seclusion of their monasteries. (Ryan and Rybolt 1995, p. 169)

Notice: this group openly states that they are *not* nuns. They do not belong to a religious order. And yet they have a Rule of Life. And yet they have a religious *vocation*. And yet they are officially recognized by the Catholic church. This rule describes their life of poverty, chastity, and obedience: not defined as

a formal monastic order, but as appropriate to women who work in the world in a common ministry together day in and day out (annual simple vows). The rule describes the love they are to express toward the sick poor and the virtues they are to exhibit in their relationships with one another. The rule specifies a set of common spiritual practices that each sister is obligated to maintain: self-examination and confession, regular assemblies for discussion, regular times of worship. Indeed, specifications are made for brief but sincere acts of devotion: hearing some devotional book read for a quarter of an hour, kneeling in silence for a moment and then reciting a prayer, setting aside brief times for meditation. What you see when you read this document is a "daily office" conducted at the workplace and wherever possible.

In founding their community in such a manner (and getting approval for this foundation), the Daughters of Charity changed the shape of the history of religious life. To quote Louise Sullivan in her introduction to the life and works of Louise de Marillac, "The form of consecrated life begun by Vincent and Louise with the Daughters of Charity has become the norm for most religious congregations. Though cloistered orders are alive and well, most religious women today live their lives in active apostolic communities" (Sullivan 1995, p. 50). I cannot emphasize how significant this is. This is a formal approval for a form of life that the beguines had been exploring three hundred years prior. Yes, there are a few nuns who pray in cloisters, but the vast majority of consecrated women today are now living lives of active service in the world, walking that tightrope of active service and trying to maintain a sincere and disciplined life of devotion and community. Diarmuid O'Murchu tells the story of a number of "paradigmatic foundresses" in his *Religious Life in the 21st Century*, many of which followed in the path of the Daughters of Charity and bear close resemblance to the Beguine Option which was—by fits and starts—increasingly permitted within the Roman Catholic community. The Daughters of Charity became the most imitated and adapted form of religious life in its time and is still today the model for many, many religious congregations, Catholic and Protestant alike.

This brings us to consider those whose ties with Rome were severed during the sixteenth century. As with other topics in this essay, I can only provide a few samples of the convoluted history of the relationship of "Protestantism" with "monasticism" (see Biot 1963; Peters 2014).

While Martin Luther and his followers dissolved the institution of monasticism within the German Protestant region, Luther himself left a window of respect open for intentional Christian communities particularly for the purpose of education (see Luther 1961, pp. 446–47). This was the window that Augustus Herman Francke (1663–1727) opened in his founding of many schools in Halle. Francke was a Pietist, a Lutheran who emphasized spiritual renewal (Erb 1983). One of the students in his school in Halle was Nicolaus Ludwig, Count Zinzendorf (1700–1760; for Zinzendorf and the Moravians see Langton 1956; Lewis 1962). In 1722, Zinzendorf offered asylum to a group of persecuted Christians on his personal estate. Soon, other religious refugees of one sort and another came to find a home. But keeping the peace among a wide range of religious refugees was not easy and things grew tense in the settlement. Zinzendorf invested himself in prayer and visitation, interviewing the members one-by-one and facilitating small group interaction as well. On 13 August 1727, they experienced a deep spiritual awakening—a true breakthrough. They drew up a covenant describing how they might continue to live together and support the Lord's work, called "The Brotherly Union and Agreement at Herrnhut" (Zinzendorf 1983, pp. 325–30). This document describes their celebration of special days of worship. It encourages but does not demand regular confession. It restricts the interaction between single persons of different sexes. It urges the community members to model virtues of the Christian life. Arrangements are made to care for the sick and to correct the erring or intransigent. There are further instructions for the "watchers," encouraging them to sing a hymn at the change of the hours of the night. In actual fact, this community instituted a round-the-clock rotation of prayer that lasted one hundred years. A wide range of mission expressions emerged from the Moravian movement. The Moravians also experienced their own tensions within the Lutheran authorities. Living in common yet not cloistered; an agreement yet no solemn vows; a combination of prayer and mission. Within the Lutheran church, two centuries after the Protestant Reformation, one can identify the emergence of

what I am calling a Beguine Option, an option that has inspired others in the centuries that followed (Winter 1994; Grieg 2004).

Anabaptists did not accept the Lutheran arrangement. They thought that reform included a change in Church–State relations and a reform of the shape of the church itself. Anabaptists found themselves collecting together in independent communities. In time, some communities documented their way of discipleship together in *Ordnungen*, an order describing community practices and behavior that were normative for a given community (see https://gameo.org/index.php?title=Congregational_ Orders_and_Church_Disciplines_(Gemeindeordnungen)). Essentially, the *Ordnungen* were rules for the life of a community living together without cloister, committed to the sharing of possessions (see Walpot 1994) without a vow of poverty. Some historians of Anabaptism see significant connections between the modern devout and the values of the Anabaptist communities (Davis 1974). Needless to say, Anabaptist forms of life have survived and are active in Mennonite, Amish, Bruderhof and other communities today.

In 1536, Henry VIII of England began what is known as "the dissolution of the monasteries." Over the next six years, hundreds of convents, friaries, houses of canons and the like were disbanded. Their assets became the property of the State, and the State made new arrangements for the thousands of religious who had previously lived in these estates. Thus, there was almost no monasticism life in England for three hundred years. I say "almost" because there were a few "Beguine-ish" exceptions. Mary Ferrar, in 1625, acquired some property in Huntingdonshire complete with an abandoned chapel. Family began to collect there and they formed a little community, complete with regular recitation of the prayers through the day and service to local children in need. The vision of the "Little Gidding" community was renewed in a community in England between 1970 and the 1990s (see Van de Weyer 1988), and their charism is currently kept alive by a new Community of Christ the Sower (http://www.stmarymagdalenes.org/christthesower.htm). The community was established for people who were not vowed religious, but rather for folks who work in the world. Members make commitments of simplicity but take no vows of poverty.

Greg Peters, in his *Reforming the Monastery: Protestant Theologies of Religious Life,* identifies a 1659 plan for an community inspired by the Carthusian order; an Anglican dean around 1668 who gave spiritual direction to twelve women "living monastically" in a convent in London; a 1707 reference to a community of women in Bath and Wells, who established an Anglican sisterhood of the Little Gidding type; and many other expressions of a longing for the renewal of some type of "Anglican" monastic life. Peters concludes his chapter on the Anglican tradition with a thorough retelling of the story of the rebirth of religious life in the context of the Oxford movement, and describes the foundation of a Sisterhood under the oversight of parish priests and which combined both contemplative and charitable dimensions (see Peters 2014, pp. 53–90).

Needless to say, many more examples can be named (see also Biot 1963; Bloesch 1964; Bloesch 1974). Bonhoffer's underground experimental seminary Finkenwalde was an exploration in semi-monastic life. George McLeod's pioneering efforts with the Iona community in Scotland combined both Christian community life and restorative manual labor. I think it is fair to say, as we follow the developments of modernity, that the religious institutions of this period increasingly made room for forms of religious life which desired to live in common without being cloistered, to live in simplicity without renouncing all property, to live pure lives without expecting members to be celibate, to receive ecclesial permission and support without becoming smothered by accountability, and to explore the combination of an intentionally devout life and occupation in the world. As we shall observe in the next section, I see a flowering of just such expressions emerging today.

## 4. Contemporary Expressions of the Beguine Option: A Pervasive Present

I began this essay with two contemporary examples of the "intentional yet less institutional" forms of life that I am here calling the Beguine Option. In section two, I described this option in greater detail, presenting a summary portrait of beguine life in the late middle ages. Then, in section three, I have traced

the "persistent past" of these forms of life, providing sufficient examples to demonstrate that Christians have often explored semi-religious ways of life through which they embody whole-hearted devotion. Having already provided a couple of contemporary examples—and there being hundreds of other examples I could give (for sample lists, see https://www.nurturingcommunities.org/communities; https://christiancommunity.org.uk/about-us/links-to-other-christian-communities/ and https://en.wikipedia.org/wiki/Society_of_apostolic_life)—it seems best to summarize the contemporary scene not by giving further examples of individual communities, but rather to summarize the work of a few relevant networks of intentional Christian expressions that bear resemblance to our Beguine Option. As we shall see in the final section, the word "network" is important as we consider what a postmodern "intentional yet not as institutional" monasticism might look like.

The Community of the Beatitudes (https://beatitudes.org/en/) was originally founded in 1973 as "The Lion of Judah and the Immolated Lamb" by the French couple Gerard and Josette Croissant and a couple of friends. They were acknowledged as an association of the faithful by diocesan right in 1985. They changed their name to Community of the Beatitudes and were received as an international association of the faithful by pontifical right (approval by the Vatican and not merely local permission) in 2002. They have long been a member of the Catholic Fraternity of Charismatic Covenant Communities and Fellowships. After some controversy and a few cases of moral failure, the Holy See intervened in 2010 and were reorganized as an "Ecclesial Family of Consecrated Life" under the Congregation for Institutes of Consecrated Life and Societies of Apostolic Life. By this time, the community of the Beatitudes numbered over seventy houses worldwide. The distinctive charism of the Community of the Beatitudes is that they welcome faithful from all states of life: married people with children, single people, consecrated lay people who live in chastity, priests, and permanent deacons whether single or married. Married and celibate singles live together in a common estate, though in designated dwellings. Associates with lesser degrees of commitment also share life together with those who make full commitment. The communities celebrate common worship, common meals, common decision-making processes, and shared service in areas of need. The Community of the Beatitudes is simply one example of the many "new communities"—clearly not individual communities, but networks of communities—established in the Roman Catholic church after the Second Vatican council.

The Fresh Expressions movement (http://freshexpressions.org.uk/) was born in 2004 as a collaboration between the Church of England and the British Methodist Church. The inspiration for the movement was some exploration into what a "mission shaped church" might look like. People began to think about the need for a mixed economy of church forms. Bishop Steven Croft, one of the early leaders, identified "a Fresh Expression is a form of church for our changing culture established primarily for the benefit if people who are not yet members of any church." A number of experiments were established and new partners have been added (the Salvation Army, the United Reformed Church, the Church of Scotland and the Baptist Union of Great Britain). Studies conducted between 2012 and 2016 showed 1109 Fresh Expressions of Church identified in 21 UK Dioceses with 50,600 attending. These statistics were just for the Church of England.

My reason for mentioning this movement is that some of the "Fresh Expressions" are monastic-like in structure and life. Indeed, in 2010 some of the leaders of the Fresh Expressions movement published a little book titled *New Monasticism as Fresh Expression of Church*. In the Introductory chapter, bishop Graham Cray, another of the pioneers of the movement, shares why he thinks New Monasticism is important to Fresh Expressions (Cray 2010). He argues that new monastic expressions (1) address the need and challenge of discipleship, (2) help Christian entrepreneurs sustain the long haul in planting church among the non-churched majority, (3) maintain the link between mission and discipleship, and (4) contribute to a 'deeper' ecclesiology. Contributors to the rest of the book share many stories of their own communities, some of which include both married and singles, living in common without being cloistered, rhythms of prayer joined with times of regular employment, commitments to simplicity without vows of absolute poverty. It is striking to see the similarities between the early beguine movement (which never really spread into England) and the monastic "fresh expressions."

The Nurturing Communities Network (https://www.nurturingcommunities.org/) is "an informal and growing network of Christ-centered intentional communities that strive to bear witness more fully to God's kingdom here on earth." (https://www.nurturingcommunities.org/vision/). Its roots go back to the flowering of Christian communities in North America during the 60s and 70s (see Jackson and Jackson 2009; Jackson 1978; Janzen 1996). Anabaptist and charismatic elements both combined as various communities found relationship and support for one another. A few forged a semi-formal alliance in Shalom Mission Communities, who provided "visitations" and helpful guidance for one another when needed (see http://www.shalommissioncommunities.org/).

In 2009, as new Christian communities were popping up all over, Jonathan Wilson-Hartgrove, then director of the School for Conversion had a conversation with David Janzen, well respected leader in the intentional Christian community scene. They both admitted the need for pastoral guidance of young communities and Janzen chose to help. David Janzen and friends began visiting communities, providing encouragement and ties between kindred spirits. This work became labeled the Nurturing Communities Project. Some of the wisdom of this work is now contained in Janzen's *The Intentional Christian Community Handbook for Idealists, Hypocrites, and Wannabe Disciples of Jesus* (Janzen 2013). Another element that developed was a series of gatherings for leaders or community folks. Regional gatherings of communities are now common.

There are probably somewhere between 50 and 100 active communities all together in this network. Nearly all are Protestant communities, with a few members of Catholic Worker houses represented at the gatherings and a strong Anabaptist element present. Intentional Christian community is the common element between the various expressions. While most of these groups do not maintain a developed office of prayer, it is interesting to note that nearly all of these communities have some pattern of common prayer or worship, often daily. Most community members are employed in regular jobs, though some work part time and spend other time volunteering. As with the early beguines, the practice of active service is important to these new communities. Various means of arranging housing, sharing of income and possessions, decision-making and other elements characterize the forms of life of these communities. The large majority of members are married or single, yet there are a handful who have made commitments to celibacy. Few have developed any kind of formal Rule of Life. Yet this is a healthy and growing network of intentional yet not necessarily institutional religious life.

The Missional Wisdom Foundation "stands are the fulcrum between spiritual tradition and innovation" (https://www.missionalwisdom.com/about/). The Missional Wisdom Foundation emerged out of the nexus of declining Methodist churches in the US, planting creative intentional communities, and theological reflection on mission and monasticism. Elaine Heath, then professor of evangelism at Perkins seminary, along with Scott Kisker, began in the 2000s to connect ideas from their Methodist heritage with insights from the history of monasticism and what was becoming known as "new monasticism." they helped foster the planting of a few intentional communities linking college students, neighborhoods, and a Methodist-based rule of life. Some of this story is recounted in Elaine Heath and Scott Kisker's book *Longing for Spring*, which specifically mentions the beguines as an historical model for consideration (see Heath and Kisker 2010). Interest in this work developed, and by 2011 Elaine Heath and friends established the Academy for Missional Wisdom. They began to train people to form "micro-communities" of faith, which Heath and Larry Duggins documented in their *Missional. Monastic. Mainline: A Guide to Starting Missional Micro-Communities in Historically Mainline Traditions* (Heath and Duggins 2014). Their work has further developed into providing resources, planting communities, teaching classes and pioneering a "dispersed community" of members located in various places linked through a common rule of life (https://www.missionalwisdom.com/rule-of-life/) and regular pattern of support and encouragement. There are about ten active communities currently connected with Missional Wisdom and more cohorts developing all the time.

The four networks I have outlined above (Community of the Beatitudes, Fresh Expressions, Nurturing Communities Network, Missional Wisdom Foundation) provide a sampling of the forms of life emerging on the edges of the Christian Church today. These networks represent the Roman

Catholic, Anglican, Anabaptist, Mainline, traditions and more. Most of these networks, though emerging from within one tradition, are openly ecumenical. And I could add more names to this list: a collection of "new friars" around the world (Bessenecker 2006, 2010), the Community of St. Anselm located at Lambeth Palace in London which provides a monastic immersion to people aged 20–35 (https://www.stanselm.org.uk/), Ravens Bread (http://www.ravensbreadministries.com/) which links together many who are lovers of solitude (new hermits?), the Order of Sustainable Faith which is forming both residential and dispersed expressions of semi-monastic life connected with the Association of Vineyard Churches (https://www.sustainablefaith.com/theorder).

My point in all of this is first to help us look at the "present" of monasticism. Looking at religious life from the perspective of these emerging networks, it is perhaps best not to speak of a demise of religious life in the twenty-first century, but rather of the early stages of a new birth of religious life. People are hungry to live an intentional—and even somewhat communal—life of devotion to God. But my second accompanying point is to note the resemblance of these new expressions to what I have traced through history as the Beguine Option. The new communities are not making solemn vows. Seldom do they abandon all property before joining, though most make some commitment to simplicity. They try to blend the active and the contemplative (though I still think that many are weak on the contemplative—more on this later). They have guidelines with varied degrees of rule-like specificity. They are largely self-governing though with valuable relationships with kindred spirits. Many of these groups are connected with networks of support and accountability such that moral failures or community conflicts can be addressed. In sum, the expressions of intentional Christian living emerging from the ground up seem to me to look much more like the beguines than the Benedictines.

## 5. The Beguine Option: A Promising Future for Religious Life Today

Having examined the persistent past and the pervasive present of the Beguine Option, it remains for us to explore the promising future of the Beguine Option. Needless to say, a full review and evaluation of the literature regarding the future of religious life, even within the Roman Catholic community, is impossible here (see for example, Schneiders 2000, 2001, 2013; Flanagan 2014; Pope 2014; Schreiter 2015; O'Murchu 2016; Maya 2018). It is sufficient to say that there is a significant reconsideration going on within the Roman Catholic community. Interpretation of the vision—and application of interpretations of the vision—of Vatican II play an important role in the reconsideration. To what degree is religious life simply part of a holiness hierarchy? To what degree are active religious a service arm of the episcopacy? How do religious balance separation from the world and authentic world engagement? How does religious life function as a prophetic witness to the Church and the world? How do we nourish hope in the midst of apparent decline? These questions and more characterize the discussions regarding the future of religious life within Roman Catholic circles. As I review this material, I am struck by the interest in forging ecumenical relationships, the interest in viewing religious life less from the perspective of institutional categories and more from dynamic functions, the openness to new and creative options.

There is very little reflection on the future of religious life from "Protestant" viewpoints, primarily because there is so little religious life in these circles. Yet some are responding to what they see as a hopeful rise of intentional Christian communities and even a new monasticism, imagining what this might (or "should") look like. Donald Bloesch advocates for a monasticism that is evangelical, a small-scale model of the church, is mixed with both men and women, acts as an agent of reconciliation between churches, is involved in outreach, is in conflict with the dominant values of the culture, and functions as an eschatological sign of the coming kingdom of God (Bloesch 1974, pp. 108–13). The contributors to *School(s) for Conversion: 12 Marks of a New Monasticism* list twelve "marks" of the monasticism they envision including "Sharing Economic Resources with Fellow Community Members and the Needy Among Us," "Humble Submission to Christ's Body, the Church," "Support for Celibate Singles Alongside Monogamous Married Couples and their Children," and Care for the Plot of God's Earth Given to Is Along with Support of Our Local Economies" (House 2005). More recently, Greg

Peters, in his *The Monkhood of All Believers: The Monastic Foundation of Christian Spirituality*, issues a restatement from the Protestant community for the universal call to holiness symbolized in our baptismal vows as a groundwork of spiritual growth pursued through ascetical practice by all believers, some of whom will find that vocation best embodied in formal religious life and others not (Peters 2018). Once again, as in Roman Catholic circles, the trend is to see potential in a devout life that is clearly intentional, yet not as institutional as traditional monasticism.

We must, of course, realize that we are at the cusp of a new era. Just as the Ancient world gave way to the medieval and the medieval gave way to the modern, so we are now in the transition between the modern and "something else." Just what that something else is, we don't know. We call the transition *postmodern*. The fact of the matter is that our institutions themselves are undergoing significant change; patterns of family, work and employment, housing, education, and church are simply not what they were seventy years ago (see Borgmann 1992; Castells 2009; Castells 2010; Castells 2011). What this means is that even trying to think of what religious life might look like *within* an institutional framework might be an exercise in futility. When Marie d'Oignies and Francis of Assisi began their experiments of life among the poor, having rejected the option of joining any of the nearby monasteries, many of their contemporaries (and perhaps Marie and Francis themselves!) regarded their form of life as non-monastic, but rather "something else." Yet it was the future of religious life. I suspect we should be prepared for something similar in our own era.

Yet whatever the future of monasticism looks like, I believe that some kind of Beguine Option has promise. Not that I entirely reject a Benedict Option (Dreher 2018). As I mentioned at the start of this essay, Alasdair MacIntyre and others have made a call for a "new Benedict." This call has inspired fresh reflections (Wilson 2010; Okholm 2007; Stock et al. 2007) and some valuable relationships (The Nurturing Communities folk met for a season at St. Johns Benedictine monastery in Collegeville where great mutual conversations were fostered). All well and good. But Benedict was a synthesizer and, in his *Rule,* brought a unifying coherence out of a vast variety of experiments, expressions and rules which preceded him. Frankly, I do not think we are ready for that kind of synthesis or unifying coherence. I am of the conviction that we are in need of just that vast variety of experiments, expressions, and rules so that down the line some new Benedict may arise an discern how the experiments can be best synthesized for the era ahead. But right now we are simply not ready. It is not that there is no place for a fresh recovery of the Benedictine charism. A Benedict Option is a worthy experiment. But my suspicion is that a Beguine option has a lot more promise in a postmodern era.

When, in 2007, I was just turning my attention from the field of Christian spirituality more generally toward monastic studies more particularly, I presented a paper to discuss among a few colleagues who called ourselves Evangelical Scholars in Christian Spirituality. I titled it "A Call to Order(s): Evangelical Monasticism in a Postmodern World—Preliminary Considerations (cf. https://spiritualityshoppe.org/a-call-to-orders-evangelical-monasticism-in-a-postmodern-world/). In this essay, I was exploring monasticism from the point of view of Protestant evangelicalism. What might monasticism in a postmodern age within this ecclesial community look like? I still think that a few of the suggestions I made in that paper are worth restating here along with a few more. As will become clear, I think that these suggestions paint a portrait of religious life which bears similarity to what I have been describing as the Beguine Option. Hence my conviction that the Beguine Option has a promising future for Christian monasticism.

First, I think that the very presence of some kind of religious life, whether more or less institutional, is itself a postmodern statement, and this would be exceptionally so in a Protestant environment which itself emerged with the modern era. The development of monasteries would be a frank admission that discipleship is much more than information transfer. Christian discipleship in a monastic setting is formation into a life and a form of life. The development of the university was characteristic of the shift from medieval to modern (Leclercq 1982). I suspect a renewal of some kind of monastic life could navigate the shift from the modern to the postmodern, just as Benedict and the Beguines helped navigate the shifts between the ancient and the medieval and between early and late medieval.

Second, I can imagine new explorers of religious life reexamining our goals and aims. After Vatican II, Roman Catholic religious began perceiving religious life less as some kind of "holiness factory" or as a functional arm of the church. Now the prophetic function is more commonly mentioned. Protestants have barely begun to consider what the aims, functions and meaning of religious life might be. All this will require a new "theologizing" of religious life (and I suspect this will be required for all traditions): examining history to discover the concerns and the questions that have driven the ups and downs of monastic history, interpreting the Scriptures and theological categories afresh to address these concerns and questions in light of the best strategies we can employ, evaluating and re-inventing practices and "the practice" of religious life for a new setting. I think that the beguines, even more than Benedict, left the door open for discovering the purpose of the religious life in the process. And while this kind of openness does not easily communicate a clear ideological frame, I think it provides what is necessary to do the work of ideological frame alignment (Wittberg 1994).

Third, I suspect that the most promising overarching community form for the future of monasticism will not be "institutions" as we think of them, but rather *networks* (or perhaps it might be better to say that many postmodern institutions will look more like networks). If the model for the modern age was the machine, I think a key model for postmodern transition is the network (Barabási 2014; Campbell and Garner 2016; Castells 2015). What this means is that our structures of leadership, our mobilization of resources and ministries, our use of property and much more will be decided more in view of links between people than by formal affiliations (see also Howard 2018, pp. 164–67). *Hubs*, like the Missional Wisdom Foundation, the Leadership Conference of Women Religious, the Nurturing Communities Network will all serve as connecting points between the various *nodes,* such as individual communities or interested persons. Once again, the image of a Beguine Option more accurately communicates this loose connection of networks within or outside or on the margin of institutions. Even the boundary marker of beguine habits expressed distinction without institutional identification, similar to other beguines yet independent to one another.

Mention of the structures of "leadership" brings us to a fourth suspicion. I suspect, with others, that Religious life must return to its origins as a movement of the laity, with leadership emerging from earned authority rather than through organizational position. The institutional elitism of the Roman Catholic Church is already being dismantled through highly publicized scandals. At the same time, the reticence of non-Catholics toward elitism will be tempered by a longing to be mentored by people who are seen to "live well." From either side, I see a growing interest in *ammas* and *abbas* and intentional communities who through lives of integrity demonstrate what impact a consecrated life could have. It is not just a matter of moving from heirarchy to a more democratic approach to leadership, but rather a growing respect for those who lead from life. Though good Benedictine models of collaborative leadership can be found, I suspect that the Beguine option presents more palatable ideology support for the kind of religious life needed in this next season.

This is not to imagine that some kind of new monasticism will be any purer or more harmonious that that of a previous age. Sexual scandals, for example, have arisen in all ages. What I hope we learn in a postmodern age is how better to deal with them. Conflicts and divisions have marred the Christian community since Acts 5–6 and I suspect that it will be no different in the future of Christian religious life. I only hope that we are able, in the midst of a postmodern monasticism, to learn wisdom regarding how better to navigate these matters.

Fifth, I continue to think that religious life is somehow caught up in identity, and that this sense of being "different" (than the world, than lay culture) will find unique expression(s) in the future of monasticism. Margaret Miles speaks of monasticism as creating a "counter-culture"; Walter Capps speaks of a "monastic impulse"; Sandra Schneiders speaks of a fundamental "God quest" undertaken by the consecrated celibate; Diarmuid O'Murchu talks about constructing a "monastic archetype"; (Miles 2000; Capps 1983; Schneiders 2013; O'Murchu 2016). I think all of these are poking at the issue of identity. A phrase that I have thrown around in the past few years: *late modern society has produced many collections, but few real communities; many individuals, but few real solitaries.* My

hope is that new forms of religious life can move beyond the late modern morass of identity (and the current postmodern confusion) into something freshly Christlike. This comes from the mutual exploration—and the somewhat consensual adoption of—practices, values, and beliefs that undergird a community's sense of who they are vis-à-vis the surrounding world. Here again, I find the Beguine Option to be a compelling model for religious life in the future. If any group knows what it means to be both caught and committed within the ambiguities of self-understanding and the pains of others' misunderstandings, it is the beguines.

Finally—and more recently—I have had this hunch that a new postmodern monasticism must be accompanied by a new postmodern mysticism. It is fascinating to me that many beguines were pioneers of the new mysticism of their own day. Some became well-known mystical writers, such as Mechtild of Magdeburg, Hadewijch, and Beatrijs of Nazareth (see McGinn 1998). I hear people like Bernadette Flanagan writing about solitude and the contemplative in her exploration of women and new monasticism or Elaine Heath talking about "the mystic way of evangelism" (Flanagan 2014; Heath 2017). Another of the 12 marks of a "new monasticism" is "Commitment to a Disciplined Contemplative Life" (House 2005). At the same time, I am keenly aware of the struggle of millennials to navigate their own faith experience. Some people have complained to me that they think new monastic communities are weak on the contemplative. Part of this is due to their own sense of the need for action. But I also think there is more. A new mysticism is needed to support the drive to new intentionally *Christian* forms of life. My suspicion is that this new mysticism will require significant permission for a variety of experience, ambiguity, and darkness. But, as I mentioned, I am only beginning to think about this. In any case, the creative mystical pioneering of the beguines can give us courage today to explore our own relationship with God in new experiences and with new language in the context of our new settings of religious life.

Thus, when I look at all of these six suggestions—presence, reexamination of goals, networks, laity and earned authority, identity, mysticism—I find the Beguine Option (intentional yet less institutional) to have significant potential in the context of a transitional, postmodern, Christianity. This Beguine Option has a persistent past, a pervasive present, and, in light of all we have seen in this final section, I think it has a promising future as well.

**Funding:** This research received no external funding.

**Conflicts of Interest:** The author declares no conflict of interest.

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
