# Peer review of "The Beguine Option: A Persistent Past and a Promising Future of Christian Monasticism"

_religions, doi:10.3390/rel10090491_

Round 1

Reviewer 1 Report

I think this is an interesting piece and of potential interest to academics, people of faith, and the broader public. I am mainly qualified to comment on the author's focus on beguines. There, I think the author must do a better job contextualizing the "Beguine Option." I understand that this is not a history journal and I assume that the author is not a historian. Nevertheless, it is impossible to assess the "Beguine Option" and its relevance to the 21st century without recognizing certain parallels historically. The essay is really all over the place with the numerous movements it mentions, even though it attempts to present them chronologically, suggesting a long history of the "option." Yet the option comes in and out of favor depending on the historical context. So I think that really has to be addressed throughout the essay. To bring things full circle, in identifying parallel options available today, the author merely says "something is going on" (line 628). This is really a very weak statement and it reflects an unfortunate tendency in the essay to describe groups without offering a substantive analysis. What is going on? I think greater attention to context will help make the author's argument more persuasive. The list of reasons the Beguine Option has appeal in the postmodern age make sense to me, but I think the argument is weak without a decent contextualization of the Beguine movement.

This requires the author to really dig into the significant social, political, and economic changes that are only glossed over in lines 97-105. I found ,myself wondering throughout the article why the author was so focused on beguines rather than some other lay religious community (why not Humiliati, Beghards, Poor Catholics?) What was special about beguines? This is never made clear. When the author states the rationale at the end (“When I look at all of these six suggestions—presence, reexamination of goals, networks, laity and 771 earned authority, identity, mysticism—I find the Beguine Option (intentional yet less institutional) to 772 have a significant potential in the context of a transitional, postmodern, Christianity), the list of suggestions DOES have medieval parallels with the beguine movement but without adequate contextualization/explanation, these parallels aren’t clear. One is left with the impression that the author selected “Beguine” randomly.

Relatedly, the author’s sweeping discussion of earlier manifestations of intentional, without institutions, doesn’t mention Francis of Assisi at all, which is very strange and weakens the argument about the varied responses to religious and social change throughout history.

Tied to this is question of context is gender. Why does the author totally ignore gender? The beguine movement was for women. The “beguine option” was the only way women were able to live lives of apostolic piety in a world that offered women few opportunities to express their spiritual aspirations. As several recent studies have shown, the beguine movement reflects major changes in medieval society in the 13th century and the institutional church’s response to it is important. As Miller (cited in the article) shows, the beguines were regarded as the quintessential laypeople. Their contributions to the spiritual life of their communities seem important to the author’s argument. But the gender component is key and I’m not sure why it isn’t addressed in the essay.

More specifically, the author would do well to read some more recent work on Beguine historiography, espeiclly Jennifer Deane’s “Beguines Reconsidered: Historiographical Problems and New Directions,” Monastic Matrix (2008) Commentaria 3461.

Laura Swan's book is a lovely popular history of the beguines, it is derived from the work of beguine historians. There isn't much in there that is original. In many cases, the author cites Swan rather than the scholarly studies on which Swan’s book is based. This is not a criticism of the author, just a suggestion to consult and cite the original studies first.

In section 2 “Medieval Beguines”, the author will find a lot of relevant and important detail on the term “beguine” in Tanya Stabler Miller, "What's in a Name? Clerical Representations of Parisian Beguines (1200-1328)." Journal of Medieval History 33, no. 1 (2007): 60-86 and more recently Labels and Libels: Naming Beguines in Northern Medieval Europe, ed. Letha Bohringer, Jennifer K. Deane, and Hildo van Engen (Turnhout: Brepols, 2014.) The latter book is essential reading for the author and much clearer on the question than Swan (who based her synthesis on these studies).

“Beguinages” are mentioned at line 97 but not defined. Simons and Stabler Miller are the best sources on the different living arrangements for beguines.

The discussion lines 106-114 seems to undermine the author’s argument and has some outdated information. The suggestion that women became beguines because there were no other options has been refuted in a number of recent studies. Stabler Miller, who the author cites and quotes demonstrates that the Beguine Option was not a “second best” option.  See also Anne Lester on Cistercian Nuns. Berman's recent book on Cistercian Nuns (the White Nuns) cautions scholars against the notion that Orders resisted providing pastoral care to women. Also, please note that Bynum (cited in this section) is referred to incorrectly as “Mary” in the bibliography. Her name is Caroline.

Line 116: 2114 à 2014

Lines 145-147: it's not clear what the author is trying to suggest with this sentence, The issue of gender is important... context is key here... beguines found support in the anti-heresy efforts of the institutional church. Later, they were themselves swept up in these accusations, mainly because they were women. It would be helpful to explain this point or, if not relevant to the focus of the essay, delete.

Starting at line 148, the author promises a description of the beguine life based on a number of studies (Swan, Miller, Amt). The only study here is Miller since Swan’s book is a popular survey based on Simons, Miller, Deane and a few others. Amt is not a study at all but a sourcebook, in this case a translation of the Ghent rule. This is fine but the author should specify that the discussion of the beguine life is based on house rules as presented in Miller and Amt.

Further, the author doesn’t really present a clear discussion of what it meant to live as a beguine. Clear discussion of this is available in Miller and (more broadly) Deane (“Beguines Reconsidered”). The latter analyzes dozens of house rules in German lands and I think the author will find this essay very helpful.

I really didn’t see the point of the section 187-92. Much of these accusations re. beguines are from hostile sources. And if the author wanted to discuss this in any substantive way, why does the author not ever mention that beguines were condemned at the Council of Vienne in the wake of the execution of the beguine Marguerite Porete? Again, this is a rather stunning omission.

Lines 210-222 are interesting but again could use some examples/historical detail. Stabler Miller's essay in Labels and Libels summarizes the essence of the beguine life from the perspective of medieval observers who argued that the beguines lived a more authentic Christian life because they did so freely (intentionally as the author says) without a rule. It seems that this is what the author is trying to get at and it would help if some medieval examples were deployed to make the point.

Line 219: I think the point needs clarifying here. It seems that the author is saying that this option was possible because the beguines shows that it's possible. This seems like a circular argument. The author might do well to contextualize this a bit better, drawing on Simon's essay "On the margins of religious life: hermits and recluses, penitents and tertiaries, beguines and beghards". Simon's work would set up the next section of the essay quite well.

The history of lay religious options is rather sweeping and, as I mentioned above, selective. By line 300, I was wondering where the friars were. Likewise I wondered if the author was going to mention the pastoral reformers at the University of Paris (James of Vitry being the best known with regard to beguines) who sought to find ways for the laity to live out their apostolic aspirations. Both Simons and Miller mention this.  Likewise, there is no reference here to lay preachers, the Waldensians, and ST. Francis, who was a lay person who sought to emulate the apostolic life.  The historical context is key. With the rise of cities, the visibility of poverty and disability, etc, people sought to live lives of service and prayer. To be in the world rather than secluded in monasteries. There is a contextual impetus here that the author is overlooking, which significantly weakens the argument.

Author Response

Just to say that I am so grateful for your suggestions. You are correct, I am not a historian and I was not aware of the latest discussions regarding the beguines (and I had just spaced Francis!). Your suggestions also helped me consider more carefully just what I was trying to accomplish with the whole piece and helped me clarify my purpose in the article. I still have so much more to explore in all this, but your suggestions have helped to make this a much better submission. Thanks so much!

Reviewer 2 Report

I am impressed by this fine study of Beguine monasticism through the ages and warmly recommend it for publication. I cannot, however, comment on the final section, which really is not my field or speciality. Speaking of the historical sections, more attention could be given to the significance of lay support to the Beguine phenomena in the later Middle Ages and especially also the negative perception of Beguines as a result of economic transformations: but a few sentences will be enough to amend this. Other than that I saw only minor lacunae in an already extensive and well researched bibliography: work by Eliana Magnani on 'house ascetics', Yaniv Fox on 'inclusive monastic societies' in Jura monasticism (which contradict the argument about the exportation of typically 'Irish' forms on p. 6), Helvétius on late antique and very early medieval forms of religious community life for women, and the following titles:

L. Böhringer, Merging into Clergy: Beguine Self-Promotion in Cologne in the Thirteenth and Fourteenth Centuries, in L. Böhringer/J. Kolpacoff Deane/H. Van Engen (ed.), Labels and Libels: Naming Beguines in Northern Medieval Europe. Turnhout 2014, p. 151-186.

L. Braguier, Servantes de Dieu: Les beatas de la couronne de Castille (1450-1600). Rennes 2019.

A. More, Fictive Orders and Feminine Religious Identities, 1200-1600. Oxford 2018.

D.I. Nieto-Isabel, Overlapping Networks. Beguins, Franciscans, and Poor Clares at the Crossroads of a Shared Spirituality, in G.T. Colesanti/B. Garí/N. Jornet-Benito (ed.), Clarisas y dominicas. Modelos de implantación, filiación, promoción y devoción en la Península Ibérica, Cerdeña, Nápoles y Sicilia. Florence 2017, p. 429-448.

These titles will help to give the argument additional depth as regards the economic and social background of the phenomenon in the later Middle Ages, resistance and persecution, and self-perception in Beguine communities. But they will not change anything that is fundamental about this paper. 

Author Response

Thank you so much for your affirmations and suggestions. I followed your bibliographic advice as best I could in the time I had for revisions. Very helpful indeed! The help from you and my other reader have made this a much better submission.

Round 2

Reviewer 1 Report

This is a much improved version and the author is to be commended for the significant work she/he did to contextualize the beguine movement. I think this background research (especially the incorporation of Deane and More's work) has done much to advance the originality and interest of this essay and I am very glad to see such a fascinating and timely application of medieval ideas to modern issues. Given recent interest in the beguines (in novels and academic publications), I think this piece will be of broad interest to academic and non-academic audiences.